# Automatic Pixel-Level Pavement Crack Recognition Using a Deep Feature Aggregation Segmentation Network with a scSE Attention Mechanism Module

**DOI:** 10.3390/s21092902

**Published:** 2021-04-21

**Authors:** Wenting Qiao, Qiangwei Liu, Xiaoguang Wu, Biao Ma, Gang Li

**Affiliations:** 1School of Highway, Chang’an University, Xi’an 710064, China; 2017021034@chd.edu.cn (W.Q.); wuxg@chd.edu.cn (X.W.); 2Inner Mongolia Transport Construction Engineering Quality Supervision Bureau, Hohhot 010020, China; 3School of Electronic and Control Engineering, Chang’an University, Xi’an 710064, China; 2018132015@chd.edu.cn (Q.L.); 2018232020@chd.edu.cn (B.M.)

**Keywords:** pavement crack detection, CrackDFANet, lightweight backbone network, scSE attention mechanism module, sub-network aggregation, sub-stage aggregation, detection speed, error rates

## Abstract

Pavement crack detection is essential for safe driving. The traditional manual crack detection method is highly subjective and time-consuming. Hence, an automatic pavement crack detection system is needed to facilitate this progress. However, this is still a challenging task due to the complex topology and large noise interference of crack images. Recently, although deep learning-based technologies have achieved breakthrough progress in crack detection, there are still some challenges, such as large parameters and low detection efficiency. Besides, most deep learning-based crack detection algorithms find it difficult to establish good balance between detection accuracy and detection speed. Inspired by the latest deep learning technology in the field of image processing, this paper proposes a novel crack detection algorithm based on the deep feature aggregation network with the spatial-channel squeeze & excitation (scSE) attention mechanism module, which calls CrackDFANet. Firstly, we cut the collected crack images into 512 × 512 pixel image blocks to establish a crack dataset. Then through iterative optimization on the training and validation sets, we obtained a crack detection model with good robustness. Finally, the CrackDFANet model verified on a total of 3516 images in five datasets with different sizes and containing different noise interferences. Experimental results show that the trained CrackDFANet has strong anti-interference ability, and has better robustness and generalization ability under the interference of light interference, parking line, water stains, plant disturbance, oil stains, and shadow conditions. Furthermore, the CrackDFANet is found to be better than other state-of-the-art algorithms with more accurate detection effect and faster detection speed. Meanwhile, our algorithm model parameters and error rates are significantly reduced.

## 1. Introduction

As an important part of the transportation hub, highways not only bear the heavy responsibility of transporting goods, but also concerns for the safety of transport personnel. However, due to natural or human factors, highways suffer from various damages. Among all kinds of pavement damage, cracks re the most common type and an early form of damage, which poses a latent threat to highway safety. To keep the pavement in good condition, it is an important task for the transportation and maintenance department to locate and repair the cracks in a timely way [1].

Manual crack detection methods are utterly dependent on the knowledge and experience of the inspection personnel, and these methods are very subjective, time-consuming, and labor-intensive. In addition, there are other crack detection techniques based on traditional image algorithms, such as edge detection [2,3] and image processing [4,5,6]. Although the crack detection methods based on traditional image algorithms have achieved better results than manual crack detection methods, they didn’t consider complex noise, and have shortcomings of low detection accuracy and detection efficiency. The deficiencies of these algorithms can be attributed to the lack of reliable feature representation and the neglect of the interdependence between cracks.

In recent years, deep learning has become a research hotspot in the artificial intelligence field. Many researchers have successfully applied these technologies to pavement crack detection. Chen et al. applied a convolutional neural network and naïve Bayes data fusion (NB-CNN) to detect cracks. The advantage of this algorithm is that it can detect thin cracks, but it can only recognize the location of cracks, and cannot accomplish the accurate extraction of cracks [7]. Dung et al. proposed a crack detection algorithm based on the fully convolutional network (FCN) for concrete crack images segmentation, which can realize the pixel-level crack detection, but the crack detection accuracy is not high due to the complex noise interference [8]. Liu et al. used the U-Net to detect the concrete cracks, because the jump layer is used in the network, the crack features can be extracted more accurately, but the robustness of the model is not good [9]. Bang et al. proposed a pixel-level detection algorithm that uses the SegNet method to detect pavement cracks, which can detect cracks at the pixel-level. However, the max-pooling operation in the down-sampling process will cause information loss. Hence, for edge cracks and noise interference cracks, the crack detection effect is not good [10]. Most of these pavement crack detection networks do not use the global context information of cracks, and lack the ability to capture important crack pixel information to guide network training. This will cause the loss of crack feature details in the process of reducing the resolution of the feature map.

The attention mechanism can refine crack features and capture important crack pixel information to guide network training. In [11], the scSE attention mechanism module that combines the channel attention mechanism [12] and the spatial mechanism [13] was proposed. The scSE attention mechanism can enhance important information features while suppressing unimportant information features in space and channels. For this reason this paper applies the scSE attention mechanism to the crack detection problem.

In mainstream crack segmentation architectures, extracting more detailed crack features can obtain better crack detection results at the output of the network. Common structures include the multi-branch framework [14], feature pyramid [15], etc. However, these structures cannot accurately obtain the contextual information of the cracks. In order to improve the learning ability of the model and increase the receptive field, this paper introduces the feature reuse idea of the deep feature aggregation network (DFANet) [16] to extract crack features from different levels.

In actual conditions, there are mainly asphalt pavements and concrete pavements, which have different pavement roughness and porosity. Asphalt pavements are rougher and the background is more complex than in concrete pavements. As shown in Figure 1, pavement crack images have the features of uneven light intensity, complex topology, low contrast, complex texture, and extensive noise interference, etc. It poses a greater challenge to deep learning-based algorithms. Meanwhile, finding a good balance between detection speed and detection accuracy has become an essential issue in the crack detection.

To solve the above challenges, this paper remodified the network architecture of DFANet, and integrated the scSE attention mechanism behind each encoder module, and proposed a novel pavement crack detection algorithm calls CrackDFANet. The CrackDFANet model integrates the advantages of the DFANet and the scSE attention mechanism, which reduces the model parameters while maintaining a good receptive field and enhancing the learning ability of the model. With all thid it can find a good balance between detection speed and detection accuracy.

For pixel-level crack detection tasks, there are only two classes: cracks and non-cracks. When crack samples are seriously insufficient in the training set, the model may incorrectly classify part of the crack samples as non-crack samples, thereby reducing the accuracy of the model. Solutions can be summarized into three categories: data-level methods, algorithm-level methods and hybrid methods [17]. To address the problem, algorithm-level methods are applied in this work. These methods can modify the algorithm to consider category penalty and weight, or modify the decision threshold to reduce the deviation of negative category samples. This paper use the focal loss to solve the class imbalance problem [18]. The main contributions of this paper can be summarized as follows:(1)This paper remodified the network architecture of DFANet, and integrated the scSE attention mechanism behind each encoder module, and the novel pavement crack detection algorithm calls CrackDFANet, which can make full use of the multi-scale receiving field to refine the crack detection results and extract the high-level features and low-level details of the crack. This allow us to achieve better results between detection speed and detection accuracy.(2)The scSE attention mechanism was integrated to the DFANet, a set of weight coefficients are learned independently through the network, and the mechanism of “dynamic weighting” to emphasize the crack region of interest while suppressing the irrelevant background region. And the attention module can correlate the global information of the crack. In this way, it effectively improves the detection efficiency of the model and reduces the computational cost.(3)The focal loss function can decrease the weight of easy-to-classify samples, and the model can focus on hard-to-classify samples during training. This paper uses the focal loss to solve the category imbalance problem.

The rest of the work is arranged as follows. Firstly, Section 2 reviews the common crack detection algorithms. Then Section 3 introduces the production process of the datasets of this paper and four common public datasets. Next, Section 4 detailly introduces the model of crack detection algorithm proposed in this paper. Section 5 carries out experimental verification and analysis of the proposed crack detection algorithm. Section 6 calculates the error rate. Finally, Section 7 summarizes the main work and conclusions of the paper.

## 2. Related Work

The pavement crack detection is essential for safe driving and transportation. For a long time, crack detection has received extensive concern by the academic and engineering circles in the world, and it has obtained some good research results. In this section, we will review and summarize some typical crack detection algorithms.

In this paper, the crack detection methods based on the non-deep learning technologies are called the traditional crack detection methods. These methods can classify into five types: (1) wavelet transform-based methods, (2) image thresholding-based methods, (3) traditional machine learning-based methods, (4) edge detection-based methods, (5) minimal path-based methods. These methods are successfully used for the crack detection, such as Gabor filters [19], histogram of oriented gradient (HOG) [20], local binary pattern (LBP) [21], and minimal path selection (MPS) [22]. Although these algorithms achieve better results than manual detection, it is essential to note that they are insufficient to differentiate between cracks and complex backgrounds in low-level images.

Recently, some deep learning technologies have been successfully used for crack detection and have achieved better detection results, such as object detection, image classification and image segmentation. The accuracy attained by the convolutional neural network (CNN) based networks has extremely surpassed the accuracy achieved by traditional crack detection methods [23,24,25]. Reference [26] used the object detection method to detect the location of cracks in pavement images. References [27,28] divided pavement crack images into smaller image blocks by using the road grid or sliding windows, and then used a CNN to recognize whether the image block includes a crack or not. In [29], the shallow CNN-based architecture for concrete crack detection was proposed. This model enabled the employment of deep learning algorithms using low-power computational devices for a hassle-free monitoring of civil structures. Le et al. proposed a crack detection algorithm based on a DL convolutional neural network, which achieved a good crack classification effect [30]. Ali et al. proposed a customized convolutional neural network for crack detection in concrete structures, which can achieve higher crack detection accuracy [31]. In [32], a two-stage data enhancement method was used to construct a CNN-based crack detector. The results show that the crack detector with two-stage data enhancement training on a small dataset has a better crack detection effect.

Although the methods above can accurately locate cracks, they cannot detect cracks at a pixel-level. For this reason, Huang et al. used the FCN for the crack detection and achieved high precision [33]. Zou et al. proposed an end-to-end deep convolutional neural network (DeepCrack) to realize the automatic detection of cracks by learning high-level characteristics of cracks. The method incorporates features of the multi-scale deep convolution learned in the hierarchical convolution stage to capture linear structures, which can achieve better detection results [34]. In reference [35], a lightweight end-to-end pixel by pixel classification network (SegNet) was used to detect cracks. In this work, SegNet used max-pooling indexes calculated in the pooling step of the encoder to implement non-linear up-sampling in the matching decoder, which doesn’t need to learn in the up-sample. The trained SegNet model can segment crack images in any size image with the help of sliding window scanning technology. Reference [36] showed that deeper backbone networks in FCN models and skip connections in U-Net both improved the performance of the crack detection. Li et al. combined generative adversarial network (GAN) with fully convolutional DenseNet (FC-DenseNet), and proposed a pavement crack detection method based on adversarial learning semi-supervised semantic segmentation. The improved algorithm can use a small number of labeled datasets to train a crack detection model with better robustness. However, when detecting relatively thin cracks, the effect is still not very satisfactory [37]. In [38], a block crack detection method was proposed, which divided the input image into block units, judged whether there were cracks in each block through the classifier, and then segmented the cracks from the classified blocks. This method can recognize the images of different sizes of cracks.

Most deep learning-based crack detection algorithms are not effective when detecting cracks with uneven intensity, complex noise interference, complex topology, etc. Meanwhile, due to the characteristics of the network themselves, the number of parameters is relatively large, and it is difficult to achieve a balance between crack detection speed and crack detection performance.

## 3. Data Collection

This section introduces the production process of datasets. To show that the algorithm proposed in this paper has good generalization ability, four public datasets are selected for the experimental analysis.

### 3.1. Our Datasets

#### 3.1.1. Data Acquisition Process

In the two types of asphalt and concrete pavement, this paper selects sunny and cloudy weather conditions to take pavement crack images at different times of the day. Compared with the concrete pavement, the asphalt pavement is rougher and the background is more complex. As shown in Figure 2, an FDR-AX60 camera (SONY, Shanghai, China) was used to take 1000 crack images with a resolution of 3840 × 2160 pixels on five highways. Then the images captured by the camera were encoded and compressed through the image acquisition card, and stored in the memory of the computer to complete the collection of crack images.

To ensure the diversity of crack images in the dataset, the crack datasets containing water stains, oil stains, parking lines, plant interference, pseudo cracks, etc. are collected at different lighting stages of the day. These crack images containing noise become training samples for network models, which can improve the robustness of the crack detection algorithm.

#### 3.1.2. Data Augmentation and Annotation

The number of crack images collected in this paper is relatively small and the size is relatively large, it is not conducive to the training of the model. In addition, the insufficient number of datasets is likely to cause overfitting in the model training. Therefore, it is necessary to use the image preprocessing to enhance the collected crack images. Through the image enhancement, the number of images can be increased. The image enhancement methods in this paper include a clockwise rotation of 90° and 180°, horizontal flip, and random color jittering. Then randomly cut into the dataset of 512×512 pixels for model training. No data augmentation is used during the testing.

In addition, this paper used the open-sourced deep learning labeling tool LabelMe [39] to manually label the crack images at the pixel level. In the manual marking process, this paper classified and marked the cracks in the image, and then saved them in the x.json file containing the image itself and related annotations. The information in the x.json file was converted into label images and label visualization, and then converted into binary label images. This paper repeated the above labeling process, and finally used the Python program to generate the corresponding label images in batches. Partial image annotation results are shown in Figure 3.

As shown in Table 1, this paper selects 3320 crack images from our crack datasets for the training and verification of the deep learning model. Crack images include asphalt cracks and concrete cracks, and this dataset is more extensive. The training set, validation set, and test set are divided at a ratio of 6:2:2, of which 2000 images in the training set and 660 images in the verification set are used to train and optimize the model, and 660 images in the test set are used to verify the effect of the model.

### 3.2. Public Datasets

In order to further verify the effect of the CrackDFANet model on the public pavement crack dataset and reflect the generalization ability of the model, this paper selects four public pavement crack datasets for the experimental verification. The labels of all public datasets use the original label images. The characteristics of the dataset are shown in Table 2.

## 4. Methods

For the purpose of obtaining more detailed characteristics of the input images to get better crack detection results in the output, we should acquire multi-layer semantic information through different layers of the convolution operation. By this means, both high-level features and low-level details can be maintained. This paper uses the concept of DFANet [16] as the basic network backbone for the pavement crack detection. Furthermore, to enhance important information features, while suppressing unimportant information features in space and channels, and effectively improve the accuracy of crack detection, we integrated the scSE module to optimize the model. Thus, the proposed network structure is called CrackDFANet. In the work of DFANet, the authors used ImageNet pretrained network model to achieve comprehensive feature extraction. Since our crack detection is a pixel-level binary classification task, and we have enough training dataset including all kinds of noise interference, so the proposed CrackDFANet does not need pretrained network models. This section first introduces the CrackDFANet in detail, and then introduces the loss function and the training process of the model.

### 4.1. Overall Architecture of the Proposed Method

In this work, the crack detection is described as a binary classification task at pixel-level. Given a crack image, the designed method will generate a crack prediction image. Figure 4 shows the structure of the proposed CrackDFANet. The model structure is explained in detail below.

As shown in Figure 4, the CrackDFANet consists of five parts: (1) the lightweight backbone network, (2) the scSE attention mechanism module, (3) the sub-network aggregation module, (4) the sub-stage aggregation module, (5) the dual-path decoder. It has been proved that the deepwise separable convolution (a depthwise convolution followed by a pointwise convolution) is one of the most effective methods for real-time reasoning. Compared with the traditional convolution, the deepwise separable convolution reduces the computation cost and the number of parameters while maintaining similar (or slightly better) performance. We alter the Xception network [42] to the backbone framework. Depthwise separable convolution is used in the Xception network, which can reduce the total amount of calculations and parameters while maintaining good performance. To pursue higher accuracy, we fuse the scSE attention mechanism module to recalibrate the output characteristics of each encoder block in both the spatial and the channel aspects. The sub-network aggregation module is centered on up-sampling the inputs from the previous backbone to the next backbone on the high-level feature graphs to refine the crack prediction results. From another point of view, the sub-network aggregation module can be viewed as a process of pixel classification from rough to fine. The sub-stage aggregation module is performed by assembling feature representations between the matching phases through “rough” part and “fine” part, which provides the receptive domain and high-dimensional framework details by connecting the layers with the same dimension. After these modules, a dual-path decoder made up of deconvolution and bilinear up-sampling operations is used to connect the outputs of each phase to obtain the crack detection results from rough to fine. The following describes the network structure in more detail:

#### 4.1.1. Lightweight Backbone Network

The primary backbone is the lightweight Xception network, which makes some changes to crack detection tasks. In the first convolution, this paper only uses ordinary convolution with a convolution kernel of 3×3, and other convolutions use the deepwise separable convolution with a convolution kernel of 3×3. Table 3 summarizes the detailed architecture of the Xception model. For crack detection, it is not only necessary to provide dense feature representation, but also how to obtain effective semantic context information is also a problem. For this reason, we fuse the scSE attention mechanism module to simultaneously recalibrate the output characteristics of each code block in terms of space and channel, which can effectively improve the accuracy of crack detection.

#### 4.1.2. The scSE Attention Mechanism Module

As shown in Figure 5, the essence of scSE module is the superposition of two attention mechanisms. After each backbone output U of the encoder enters the scSE module, it will enter two branches, the upper branch is called sSE branch. Through a 1×1 convolution operation, the branch obtains a weight matrix with the same length and width as the U. the matrix is multiplied with U to obtain the feature $U_{\mathrm{sSE}}$ that is recalibrated in space. The following branch is called cSE branch. This branch first passes a maximum pooling operation to obtain a weight matrix with the same number of channels as the U. This matrix passes through two fully connected (FC) layers, the number of neurons in the first FC layer is half the number of channels (to reduce computational complexity). The number of the second FC layer is equal to the number of channels. After the activation function, the nonlinearity is added, and the matrix restored to the number of channels through the second FC layer is multiplied by U to obtain the characteristic $U_{\mathrm{cSE}}$ recalibrated in the channel direction. Finally, merge the features recalibrated along the channel and the space to output $U_{scSE}$. The scSE module can effectively enhance important information features and suppress unimportant information features in both the space and the channel, and can effectively improve the accuracy of crack detection.

#### 4.1.3. Sub-Network Aggregation Module

Sub-network aggregation accomplishes the association of high-level features at the network level. In this paper, our crack detection network architecture consists of a series of backbone networks, which provide the output of the prior backbone network to the next as input. From another point of view, sub-network aggregation can be viewed as a refining process. The backbone process is expressed as y=ϕ(x), and the output of encoder$\;\phi_{n}$ is the input of the encoder$\;\phi_{n+1}$. Therefore, the sub-network aggregate process can be expressed as:(1)$$Y=\phi_{n}\left(\phi_{n-1}\left(\cdots \left(\phi_{1}\left(X\right)\right)\right)\right)$$

#### 4.1.4. Sub-Stage Aggregation Module

The focus of sub-stage aggregation is to fuse semantic and spatial information at the stage level among multiple networks. With the increase of network depth, the spatial details are gradually lost. Common methods, such as the U-shape, implement skipping connections to restore image details in the decoder module. Nevertheless, due to the lack of low-level features and spatial information in deeper encoder blocks, so they cannot judge a wide variety of objects and precise structural edges. The parallel branch design usages the original and reduced resolutions as input, and the output is the fusion of the results of large-scale and small-scale branches. However, this design lacks information interaction between parallel branches.

In this paper, sub-stage aggregation is used to combine features through the encoding period. Under the same depth of the sub-network, the different stages are fused. In more detail, the output of one stage in the prior sub-network contributes to the input of the matching stage position of the next sub-network.

For a unitary backbone network$\;\phi_{n}\left(x\right)$, a stage process can be defined as$\;\phi_{n}^{i}$. The previous stage in the backbone network is $\phi_{n-1}^{i}$. i is the index of the stage. The sub-stage aggregation approach can be expressed as:(2)$$x_{n}^{i}=\{\begin{array}{c} x_{n}^{i-1}+\phi_{n}^{i}\left(x_{n}^{i-1}\right),\;if\;n=1\;\\ \left[x_{n}^{i-1},x_{n-1}^{i}\right]+\phi_{n}^{i}\left(\left[x_{n}^{i-1},x_{n-1}^{i}\right]\right),\;otherwise\; \end{array}\;$$
where $x_{n}^{i-1}$ is coming from:(3)$$x_{n}^{i-1}=x_{n-1}^{i-1}+\phi_{n-1}^{i}\left(x_{n-1}^{i-1}\right)$$

In our algorithm, the stage aggregation approach is to learn the residual formula of $\left[x_{n}^{i-1},x_{n-1}^{i}\right]$ at the beginning of each stage. In case of n>1, the input of the ith stage of the nth network is obtained by the output of the ith stage of the (n−1)th network and the output of the (i−1)th stage of the nth network, and then the ith stage learns a residual representation of $\left[x_{n}^{i-1},x_{n-1}^{i}\right]$. $x_{n}^{i-1}$and $x_{n-1}^{i}$ have the same resolution, and we achieve the fusion characteristics of concatenation operation.

We always keep the features flowing from high resolution to low resolution. Equation (2) not only learns the n−th feature maps, but also retains (n−1)−th features and receptive field. Meanwhile, Information flows can be transmitted over multiple networks.

#### 4.1.5. The Dual-Path Decoder

After the crack feature is extracted by the encoder network, the corresponding decoder network needs to be used to restore the feature map to a binary image with the same size as the input image, so as to obtain the detection result of the crack. To improve the real-time reasoning ability of the CrackDFANet model, the decoder network in this paper is designed as an efficient feature up-sampling module, which combines the crack features from the low-level and high-level of the image.

In the decoder network of CrackDFANet, this paper uses convolution and bilinear up-sampling to directly fuse the high-level features and low-level details of the crack to form a simple decoder network. Since the encoder network of CrackDFANet is composed of three backbone networks, this paper first merges high-level features from the bottom of the three backbone networks, and then performs 4× bilinear up-sampling on these high-level features, and fuse them with the same spatial resolution. The low-level information of each backbone network is finally added with high-level features and low-level details, and 4× bilinear up-sampling is performed to obtain the final crack detection result. In the decoder module, to decrease the number of channels, a small number of convolution operations are used.

### 4.2. Loss Function

In the deep learning model, the loss function is the most crucial mathematic constituent part. The loss function is applied to measure the deviation between the predicted value and the true value [43]. Therefore, the loss function is usually defined as the objective function of the crack detection model optimization. What we should pay attention to is that compared with the non-crack pixels, crack pixels occupy a small part of the crack image. Hence, the definition of total loss may be affected by this class imbalance between the two categories of cracks and non-cracks. For this reason, this paper uses a novel loss function: focal loss (FL) solves the class imbalance problem [18]. The FL is modified on the basis of the standard cross-entropy loss. And the FL function can decrease the weight of easy-to-classify samples. Therefore, the model can center on hard-to-classify samples during training. The FL can be expressed as:(4)$$FL\left(p_{t}\right)=-\alpha_{t}{\left(1-p_{t}\right)}^{\gamma }\log\left(p_{t}\right)$$where the weight factor $\alpha_{t}=0.25$,$\;{\left(1-p_{t}\right)}^{\gamma }$ represents the modulation coefficient. $\;{\left(1-p_{t}\right)}^{\gamma }$ is added, the loss obtained by the sample with a large prediction probability will be reduced, and the loss of the sample with a small prediction probability will become large, thereby strengthening the attention to the positive sample. The $p_{t}$ is defined as:(5)$$p_{t}=\{\begin{array}{c} p,\;y=1\\ 1-p,\;y=0 \end{array}$$
where y∈{0,1} is the ground truth class, p∈{0,1} is the model’s estimated probability for label y=1. t∈{0,1} represents the category number, t=0 denotes the background pixel, and t=1 denotes the crack pixel.

The properties of focal loss are as follows:

(1)When an example is misclassified. At this time, $p_{t}$ is small, and then the modulation factor $\left(1-p_{t}\right)$ is close to 1, and the loss is not affected. At this time, the model enhances the focus on positive samples. When $p_{t}\to 1$, the modulation factor $\left(1-p_{t}\right)$ is close to 0, which indicates that the classification is true and the samples are easy to classify at this time, and the modulation coefficient$\;{\left(1-p_{t}\right)}^{\gamma }$ will approach 0, that is, the sample contributes little to the total loss.(2)The focusing parameter γ(γ≥0) smoothly adjusts the proportion of the weight of the easy-to-divide samples to be reduced. Increasing γ can enhance the influence of modulation factors. γ=2 in this paper.

### 4.3. Model Training and Optimization

To obtain a more robust crack segmentation model, this paper selects 2000 crack datasets, including water stains, oil stains, parking lines, plant disturbances, rough backgrounds, etc. as the training set, and 660 datasets as the validation set for the model training. These datasets contain asphalt crack images and concrete crack images. The training structure diagram is shown in Figure 6.

Meanwhile, this paper applies the Adam algorithm to optimize the model during model training. The Adam algorithm can calculate the adaptive learning rate of different parameters from the first-order and the second-order estimation of the loss function. In particular, the Adam algorithm combines the advantages of two stochastic gradient descent algorithms. The first is the AdaGrad algorithm, which adopts a specific learning rate for each parameter to improve the performance of the spare gradient. The second is the RMSProp algorithm, which adaptively assigns a learning rate to each parameter based on the average of the closest magnitude of the weighted gradient [44].

During the model training process, the model is saved every 10 epochs, and then each saved model is tested, as shown in Figure 7. The experimental results show that the model has the best detection effect when it is trained for 600 epochs. After training, the model appears over-fitting phenomenon, and the results of crack model detection become worse. Therefore, this paper chooses to train 600 epochs.

As shown in Figure 8, through 600 epochs iteration, a real-time crack segmentation model with good robustness is obtained. In this time, the loss of model training converges to 0.024.

## 5. Experiment and Analysis

The previous section introduced the crack detection algorithm proposed in this paper. This section will verify and discuss the proposed algorithm through experiments. Firstly, we introduce the implementation details of the proposed CrackDFANet. Then we explain the evaluation criteria. Next, we introduce the compared methods. Finally, we analyze and discuss the experimental results.

### 5.1. Implementation Details

#### 5.1.1. Computation Platform

All algorithms in this paper used Python 3 for programming. The algorithms were implemented by using the deep learning framework Pytorch. The experimental software environment is Windows 10 operating system, and the hardware environment is an Intel^®^ Quad-CoreTM i7-9750 CPU @ 3.6 GHz Processor, 16 GB RAM and an NVIDIA GeForce GTX 1080Ti 16 GB GPU.

#### 5.1.2. Parameter Settings

The hyperparameters include: the momentum is set to 0.9, and the weight decay is set to 0.0001. The mini-batch stochastic gradient descent (MSGD) with a batch size of 8 is used to update the network parameters. As a general configuration, the “poly” learning rate strategy is applied where the initial rate is multiplied by ${\left(1-\frac{iter}{\max\_\mathrm{iter}}\right)}^{power}$ with power 0.9, and the base learning rate is set as 0.001 [16].

### 5.2. Evaluation Criteria

For the crack segmentation task, the generally applied evaluation indexes are: precision (P), recall (R), F1, accuracy (ACC) and mean intersection over union (MIoU). Crack pixels are defined as positive instances, while non-crack pixels are defined as false instances. According to the combination of ground truth and prediction, pixels are divided into four types: true positive (TP), false positive (FP), true negative (TN) and false negative (FN), as shown in Table 4.

Then, P, R, F1, ACC and MIoU can be defined as follows:(6)$$P=\frac{TP}{TP+FP}$$
(7)$$R=\frac{TP}{TP+FN}$$
(8)$$F1=\frac{2\times P\times R}{P+R}$$
(9)$$ACC=\frac{TP+TN}{TP+FN+TN+FP}$$
(10)$$MIoU=\frac{1}{2}\left(\frac{TP}{TP+FN+FP}+\frac{TN}{TN+FP+FN}\right)$$

### 5.3. Compared Methods

We compare the performance of CrackDFANet with FCN, SegNet, and U-Net crack detection algorithms. Meanwhile, to solve the problem of class imbalance, the loss function of all comparison methods in this paper chooses the focal loss. The following is a brief description of the methods compared:

(1)FCN: We use FCN-8s to detect cracks by replacing the loss function with FL. The hyperparameter settings are as follows: the base learning rate is set to $1\times 10^{-5}$, the momentum is set to 0.99, and the weight decay is set to 0.0005. We train the FCN model on our dataset.(2)SegNet: This network can achieve end-to-end learning and crack segmentation by using an encoder network and its corresponding decoder network. Except that the basic learning rate is set to $1\times 10^{-6}$, the settings of other hyperparameters are the same as those in FCN. We train the SegNet model on our dataset.(3)U-Net: This network uses skip-layer in the encoder-decoder network for crack segmentation. Except that the basic learning rate is set to $1\times 10^{-3}$, the settings of other hyperparameters are the same as those in FCN. We train the U-Net model on our dataset.(4)CrackDFANet: The CrackDFANet model is trained on our dataset.

### 5.4. Experiment Results and Discussion

To verify the crack detection effect of the trained and validated model described in Section 4, we compared and analyzed FCN, SegNet, and U-Net crack detection algorithms on our dataset, GAPs384 dataset, Crack500 dataset, AigleRN dataset and CFD dataset, respectively. The following is a comparative analysis of the experimental results on five datasets. In addition, to further compare and analyze the advantages of the proposed crack detection method, we have carried out experiments in some special cases, that is, light interference, water stains, oil pollution, and shadow, etc. During the test phase, the hyperparameter settings of all algorithms are the same as the training process.

#### 5.4.1. Results on Our Dataset

On our dataset, we perform the crack detection effect verification. As shown in Figure 9, the crack detection results of five typical input images of this method and the comparison method are given. The first column is the original crack image, the second column is the label image, and the next four columns are the output images of the comparison detection algorithm. The first four rows of images are concrete cracks, the latter one is asphalt cracks.

It can be seen from Figure 9 that all these algorithms can detect the crack region. However, for the thin crack and the crack image with noises, CrackDFANet obtains crack detection results close to the label image, while the other algorithms produce many false detections.

As shown in Table 5, we made a quantitative comparison of several deep learning-based crack detection algorithms. The MIoU of our crack segmentation algorithm is 0.8972, which is significantly higher than the other three crack segmentation algorithms. Meanwhile, our model parameter is 14.5 M, which is significantly reduced compared to other algorithms. The speed is an important factor in crack segmentation algorithms, and we try to compare crack segmentation models in the same state. In this paper, the speed of crack detection is measured by the frames per second (FPS). The detection speed of our crack segmentation algorithm reaches 64.5 FPS with 512×512 pixels. In other words, our algorithm takes 15.50 milliseconds to test an image. It can be seen that our algorithm has achieved a good balance between detection accuracy and detection speed. One way to obtain the model speed of the model is to summarize its calculations. We use floating point operations per second (FLOPs) to measure the computation speed of the model. From the experimental results in Table 5, it can be concluded that the calculation speed of our model is much faster than the speed of the other three comparison models.

#### 5.4.2. Results on GAPs384 Dataset

To further verify the generalization ability of the CrackDFANet, we performed experimental verification on the public GAPs384 dataset (1920×1080 pixels). The crack images in this dataset are all asphalt cracks. Figure 10 shows the five most representative crack detection effects of different algorithms under the same conditions.

As shown in Figure 10, the topological structure of the images in the GAPs384 dataset is complex, and noise interference is massive, which poses greater challenges to various crack detection algorithms. Under this kind of complicated noise interference, our algorithm still shows good detection performance, and FCN, SegNet, and U-Net crack detection algorithms are very willing to erroneously detect, such as judging parking lines, oil pollution, and manhole covers as crack regions. This is because our detection algorithm recognizes crack characteristics through the sub-network aggregation and the sub-stage aggregation. Sub-network aggregation can be regarded as a rough-to-fine process of the pixel classification. Sub-stage aggregation is mainly the fusion of semantic and spatial information at the stage level from multiple networks. Meanwhile, we fuse the scSE attention mechanism module to simultaneously recalibrate the output characteristics of each code block in terms of space and channel. In this way, the crack characteristics can be refined multiple times, thereby achieving better detection results.

In Table 6, we performed a quantitative comparison on the GAPs384 dataset. Our algorithm has an MIoU of 0.8723, and a detection speed is 8.2 FPS at 1920×1080 pixels, which is higher than the other three deep learning-based crack detection algorithms under the same conditions.

#### 5.4.3. Results on Crack500 Dataset

In addition, to further compare the detection performance of the algorithms, we carried out experimental verification on a public Crack500 dataset with larger image size (2000×1500 pixels). The crack images in this dataset are all asphalt cracks. Figure 11 shows five representative visual comparison results.

As shown in Figure 11, the texture of the cracks in the Crack500 dataset is complex, the contrast is low, and the characteristics of the cracks are not obvious. It is very apt to cause false detection. However, the experimental results show that in this complicated case, our crack detection algorithm still shows a good detection effect, in contrast, other compared algorithms have poor anti-interference ability, which shows that the CrackDFANet has better robustness and better detection effect under the same conditions.

To quantitatively compare the detection performance of various algorithms, we compute P, R, F1, ACC, MIoU and FPS on the Crack500 dataset. Table 7 shows the quantitative results. The results show that the CrackDFANet realizes best P, R, F1, ACC, MIoU and FPS and surpasses three other algorithms. This proves that the CrackDFANet has better generality than other comparison algorithms. Meanwhile, it shows that our detection algorithm improves the detection effect while increasing the detection speed.

#### 5.4.4. Results on AigleRN Dataset

In this set of comparative experiments, we conduct experimental analysis on the public AigleRN dataset, the size of the dataset is 768×512 pixels. The crack images in this dataset are all asphalt cracks, and cracks are generally thin and have low contrast, which is a challenging task for all detection algorithms.

The results of the AigleRN dataset are shown in Figure 12. Although FCN, SegNet, and U-Net can detect most the cracks in the image, but it still retains a lot of noises. In addition, due to the loss of information, some of the cracks detected are discontinuous. Meanwhile, they may hallucinate cracks that do not exist. Under the same conditions, the CrackDFANet can extract the edges of cracks in images, whether they contain some types of noise or not. This proves that our algorithm has better high interference capability and better detection effect.

The quantitative results of the experiment are shown in Table 8, CrackDFANet shows good results on most indicators. Specifically, the CrackDFANet demonstrates not only a higher detection accuracy but also faster detection speed, and it has better engineering application value.

#### 5.4.5. Results on CFD Dataset

In this set of comparative experiments, we selected the public CFD dataset for experimental verification, the size of the dataset is 480×320 pixels. This dataset contains asphalt cracks and concrete cracks.

To further verify the generalization ability of the method, four methods are tested on CFD dataset. Results are shown in Figure 13. The noise interference of the CFD dataset is relatively small, and all methods can roughly detect the contour of the crack. It can be intuitively observed that our method is superior to other methods. Three other comparisons of crack detection algorithms based on deep learning do not perform well on edge cracks and web cracks. They may hallucinate cracks that do not exist. Through comparative analysis, our proposed algorithm still achieves good detection results. The predicted image is basically close to the label image.

As shown in Table 9, compared with other crack detection methods, the CrackDFANet achieves the highest P, R, F1, ACC, MIoU, and FPS. The MIoU value of CrackDFANet reaches the highest 87.59%. The FPS of CrackDFANet is 111.3. In other words, it takes 8.98 milliseconds to detect an image. The performance improvement is mainly due to the use of the sub-network aggregation and the sub-stage aggregation. In addition, we fuse the scSE attention mechanism module to the lightweight Xception network, which can simultaneously recalibrate the output characteristics of each code block in terms of space and channel. In this way, the crack characteristics can be refined multiple times, thereby achieving better detection results.

#### 5.4.6. Special Cases Discussion

To further compare and analyze the CrackDFANet and other deep learning-based several crack detection algorithms, we perform experiments in some special cases, i.e., light interference, parking line, water stains, oil stains, and shadow, etc. Figure 14 shows five representative results showing all methods. The second and third lines are asphalt crack images, and the rest are concrete crack images.

As shown in Figure 14, under the interference of complex noise, real cracks are surrounded by disturbing fake cracks. In this case, because the patterns between cracks and pseudo-cracks are similar, which is a challenging problem for all crack detection algorithms. Compared to the other crack detection algorithms, the CrackDFANet can produce clearer and better results, regardless of whether they contain some noises or not. For crack only images, light interference images, parking line images, water stains, plant disturbance images, oil stains, and shadow images, the CrackDFANet can accurately extract crack edges, and it is best to eliminate different types of noise. This shows that the CrackDFANet model is more resistant to noise interferes than FCN, SegNet and U-Net crack detection algorithms.

In addition, to further prove the performance of different methods, we draw the precision-recall curve (PR-curve) based on the precision and recall of various comparison methods on our dataset, GAPs384 dataset, Crack500 dataset, AigleRN dataset, and CFD dataset. As shown in Figure 15, CrackDFANet is the highest among other three comparison methods, which shows that the CrackDFANet has better crack detection effect and generalization ability.

## 6. Error Rate

To compare the accuracy of different methods in crack detection, the error rate is introduced, that is, the ratio of the number of misclassified pixels to the total number of pixels in the image. The formula description is shown in Equation (11):(11)$$Error\;rate=\frac{N_{error}}{N_{total}}\times 100\%$$
where $N_{error}$ denotes the number of misclassification pixels, $N_{total}$ denotes the total pixel number of the images. The error rates of four algorithms are shown in Table 10.

The error rate of the CrackDFANet model on average is 0.73%, while for FCN, SegNet, and U-Net, it is 5.75%, 6.35%, and 4.83%, respectively. That is to say, the error rate of the CrackDFANet is obviously lower than that of the other three comparison methods. The CrackDFANet recognizes crack characteristics through the sub-network aggregation and the sub-stage aggregation. In this way, the crack characteristics can be refined multiple times. In addition, to pursue higher accuracy, we fuse the scSE attention mechanism module to recalibrate the output characteristics of each encoder block in both the spatial and the channel aspects. Based on the above analysis, the CrackDFANet algorithm can significantly reduce the error rate.

## 7. Conclusions

In this work, the CrackDFANet is proposed for the pavement crack detection. The main work and conclusions are as follows:

(1)Model validation performed on five different crack datasets, CrackDFANet records the MIoU of 0.8972 on our dataset. And the model processes in real-time (64.5 FPS) images with 512×512 pixels. At the same time, to further verify the generalization ability of the model, the model tests were performed on four public datasets: GAPs384, Crack500, AigleRN and CFD. Experimental results show that CrackDFANet achieves best P, R, F1, ACC, MIoU and FPS and surpasses FCN, SegNet, and U-Net crack detection algorithms.(2)The CrackDFANet can get a good balance between detection accuracy and detection speed. Meanwhile, the parameters of the model are greatly reduced.(3)Under the interference of light interference, parking line, water stains, plant disturbance, oil stains, and shadow conditions, the CrackDFANet has strong anti-interference ability, and has better robustness and generalization ability.(4)To compare the accuracy of different algorithms in the crack detection, we calculated the error rate for different images. The experimental results show that the average error rate of our algorithm is 0.73%, which is significantly better than the other three comparison algorithms.

In future research, we will build a wider crack dataset and test our crack detection algorithm on more datasets.

## Figures and Tables

**Figure 1 sensors-21-02902-f001:**
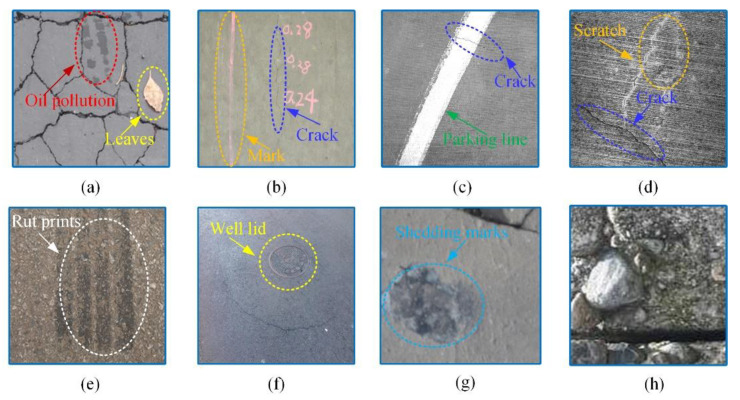
Pavement crack datasets under the interference of some complex noises: (**a**) oil pollution and leaves, (**b**) mark, (**c**) parking line, (**d**) scratch, (**e**) rut prints, (**f**) wall lid, (**g**) shedding marks, (**h**) complex background.

**Figure 2 sensors-21-02902-f002:**
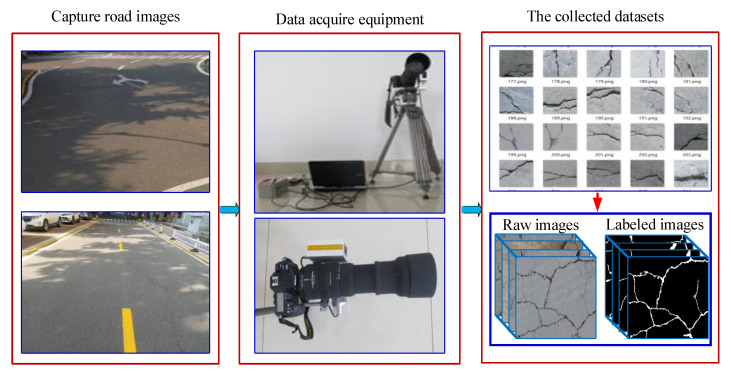
Crack image acquisition process.

**Figure 3 sensors-21-02902-f003:**
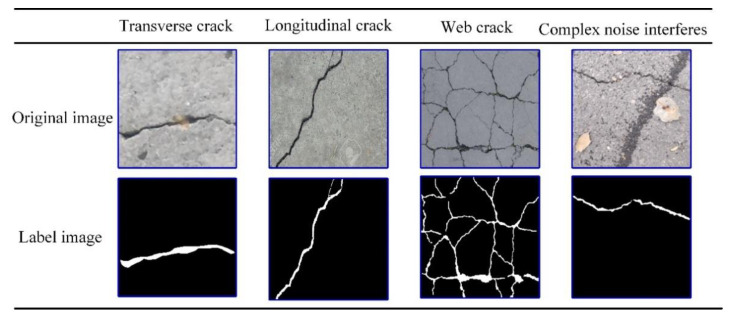
Examples of partially labeled images.

**Figure 4 sensors-21-02902-f004:**
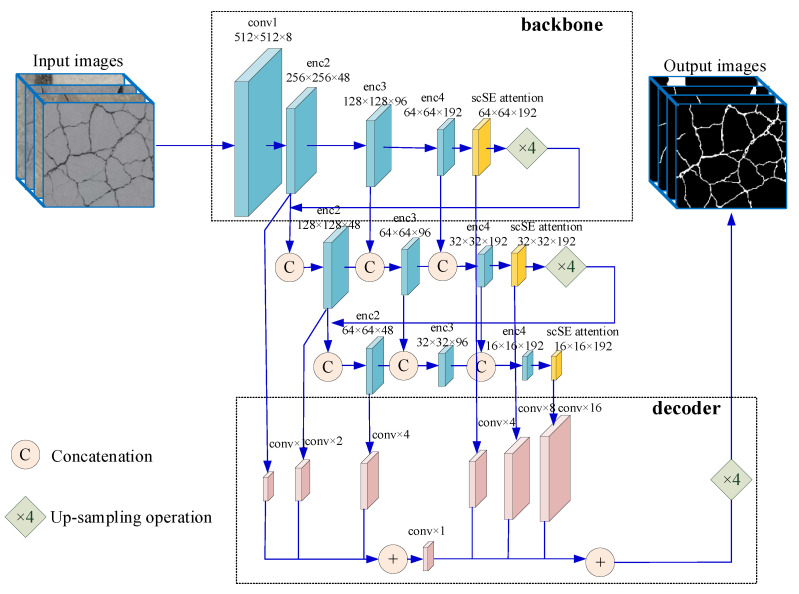
The structure of CrackDFANet: the lightweight Xception network, scSE attention mechanism model, sub-network aggregation, sub-stage aggregation, and dual-path decoder for multi-level feature fusion.

**Figure 5 sensors-21-02902-f005:**
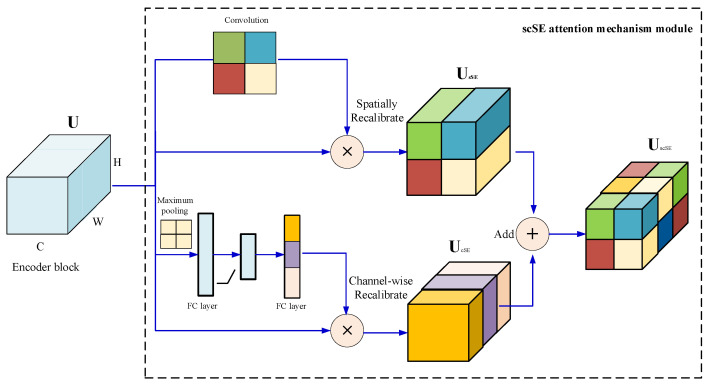
scSE module structure.

**Figure 6 sensors-21-02902-f006:**
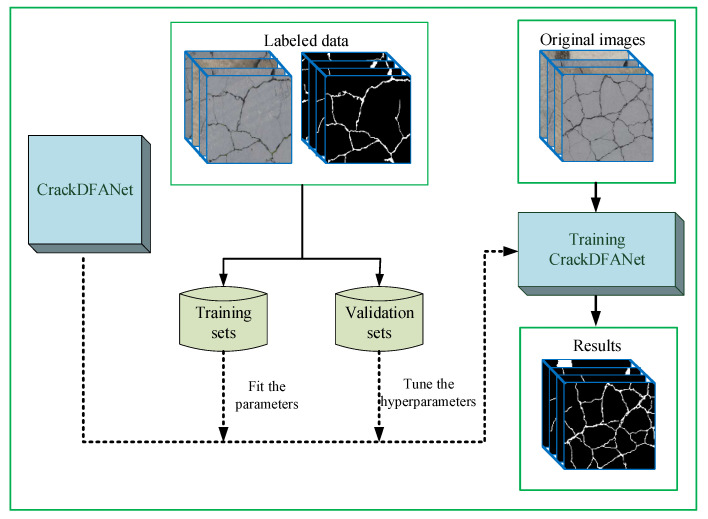
A schematic of CrackDFANet trained for concrete crack detection.

**Figure 7 sensors-21-02902-f007:**
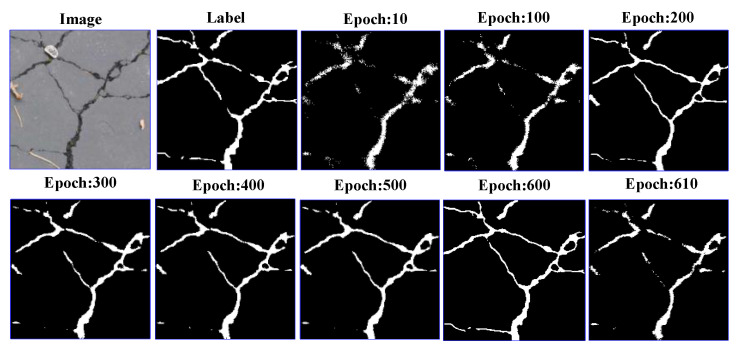
Output at different epochs.

**Figure 8 sensors-21-02902-f008:**
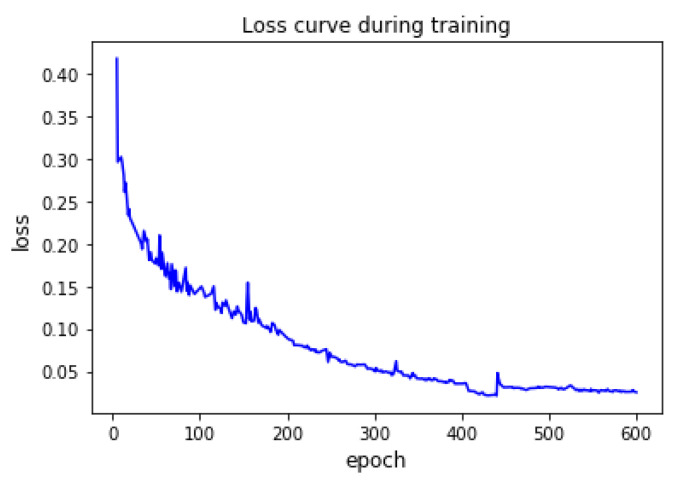
Loss curve during training iterations of CrackDFANet.

**Figure 9 sensors-21-02902-f009:**
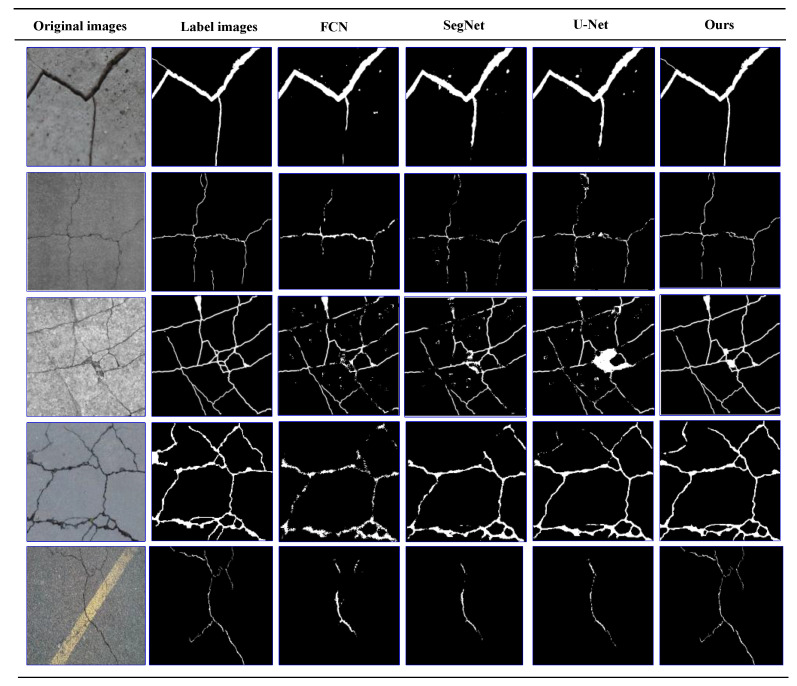
The visualization of detection results of compared methods on our dataset (from left to right: original images, label images, FCN, SegNet, U-Net, our detection results).

**Figure 10 sensors-21-02902-f010:**
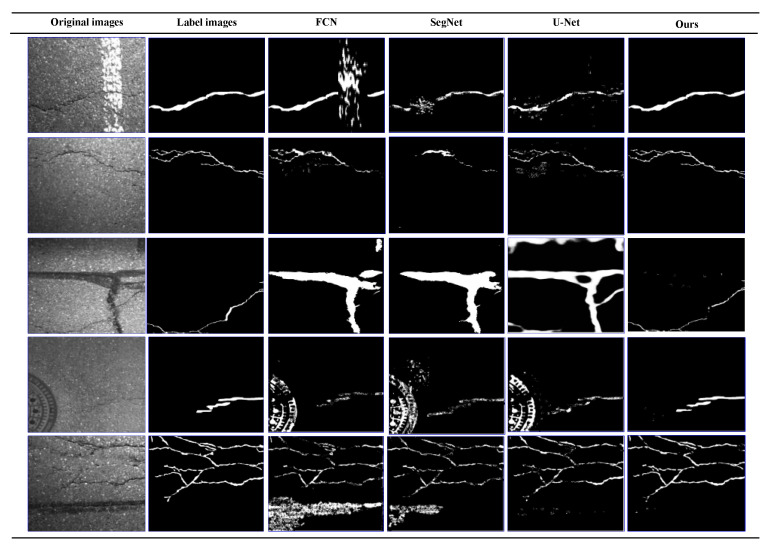
The visualization of detection results of compared methods on GAPs384 dataset (from left to right: original images, label images, FCN, SegNet, U-Net, our detection results).

**Figure 11 sensors-21-02902-f011:**
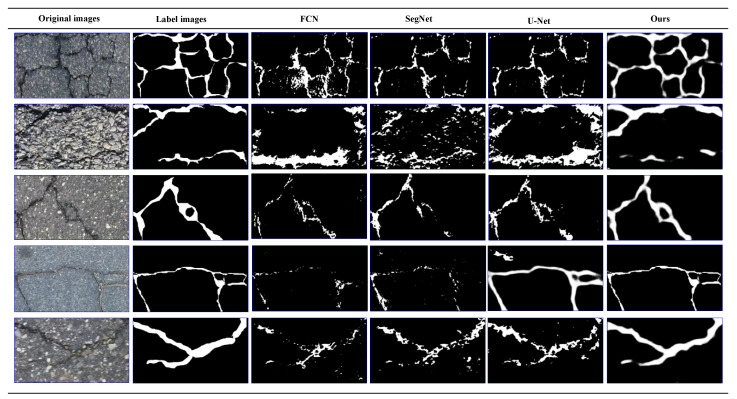
The visualization of detection results of compared methods on Crack500 dataset (from left to right: original images, label images, FCN, SegNet, U-Net, our detection results).

**Figure 12 sensors-21-02902-f012:**
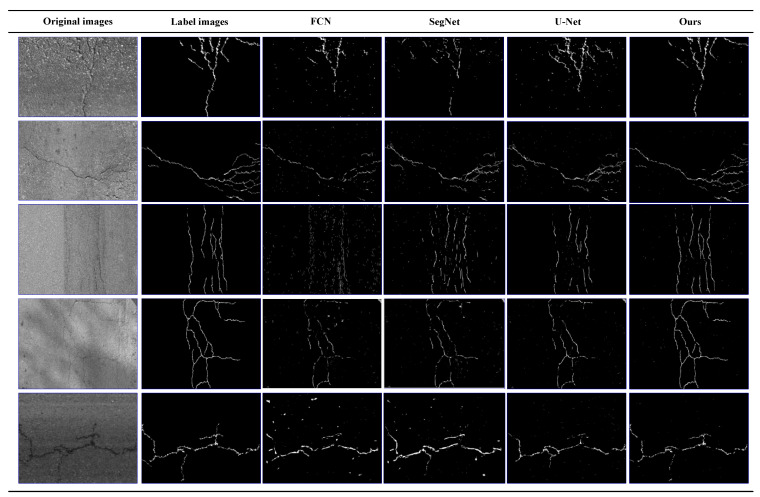
The visualization of detection results of compared methods on AigleRN dataset (from left to right: original images, label images, FCN, SegNet, U-Net, our detection results).

**Figure 13 sensors-21-02902-f013:**
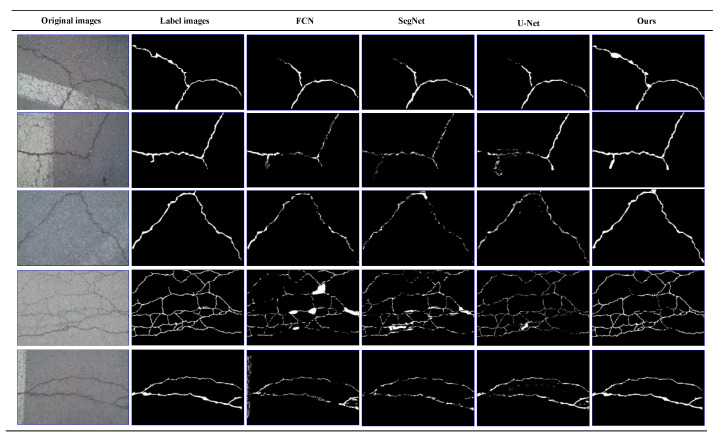
The visualization of detection results of compared methods on CFD dataset (from left to right: original images, label images, FCN, SegNet, U-Net, our detection results).

**Figure 14 sensors-21-02902-f014:**
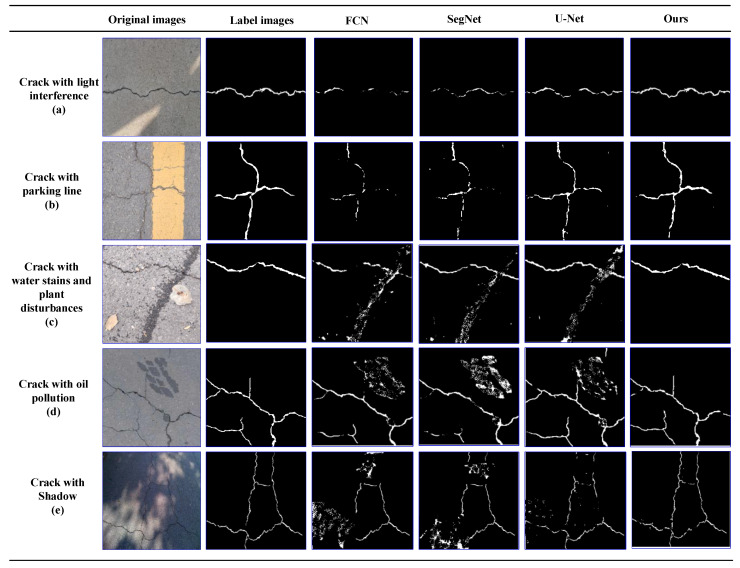
The visualization of detection results of compared methods on special cases, i.e., light interference, parking line, water stains, oil pollution, and shadow, etc. (from left to right: original images, label images, FCN, SegNet, U-Net, our detection results).

**Figure 15 sensors-21-02902-f015:**
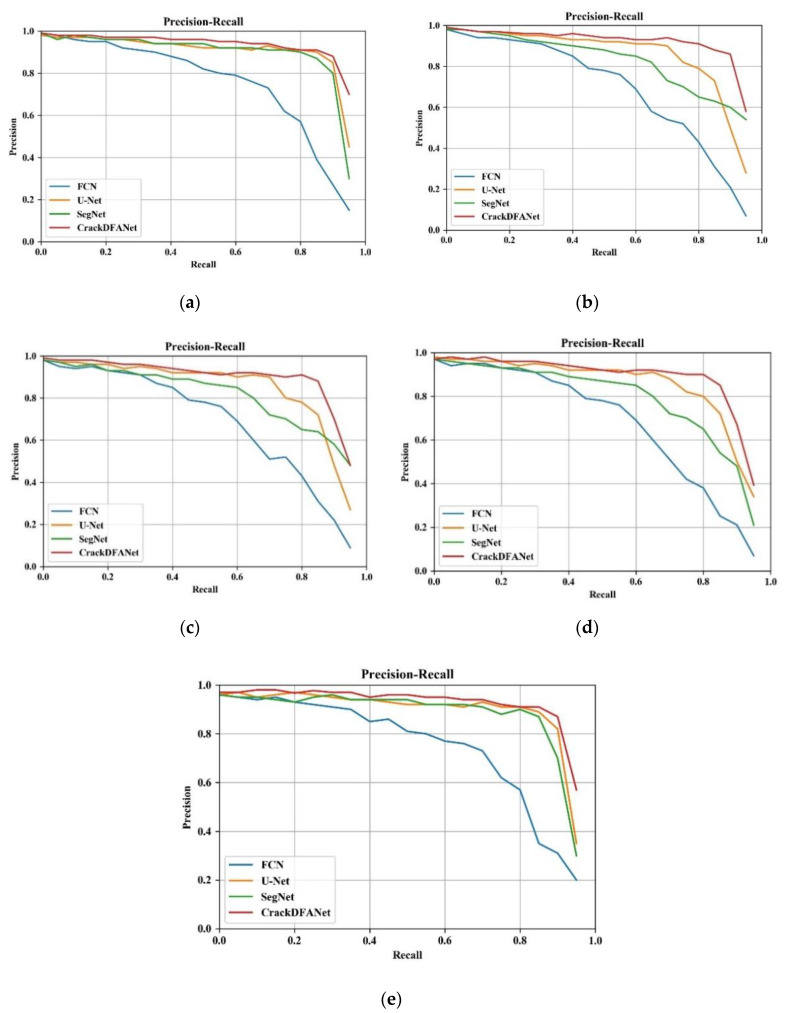
Precision-recall curve on three group experiments: (**a**) our dataset, (**b**) GAPs384 dataset, (**c**) Crack500 dataset, (**d**) AigleRN dataset, (**e**) CFD dataset.

**Table 1 sensors-21-02902-t001:** Crack image datasets details table.

Image Description	Training Sets	Verification Sets	Test Sets
Image size(pixels)	512×512	512×512	512×512
Number of images	2000	660	660

**Table 2 sensors-21-02902-t002:** The characteristics of the public crack dataset.

Datasets	Image Size (Pixels)	Number of Images
GAPs384 [40]	1920×1080	1969
Crack500 [41]	2000×1500	500
AigleRN [22]	768×512	269
CFD [1]	480×320	118

**Table 3 sensors-21-02902-t003:** The detailed architecture of the Xception model (3×3 means a depthwise separable convolution except “conv1”. In “conv1” stage, we only implement a 3×3 convolution layer).

Stage	Kernel Size	Padding	Stride
conv1	3×3	8	2
enc2	3×3	12	4
	3×3	12	4
	3×3	48	4
enc3	3×3	24	6
	3×3	24	6
	3×3	96	6
enc4	3×3	48	4
	3×3	48	4
	3×3	192	4

**Table 4 sensors-21-02902-t004:** All the results of ground truth case and predicted case.

	Ground Truth Case	Crack	Non-Crack
Predicted Case			
Crack		True positive (TP)	False positive (FP)
Non-crack		False negative (FN)	True negative (TN)

**Table 5 sensors-21-02902-t005:** The P, R, F1, ACC, MIoU and FPS of compared methods on our dataset (512 × 512 pixels).

Methods	Detection Effect Evaluation Index							
	P	R	F1	ACC	MIoU	Params	FLOPs	FPS (Milliseconds/Image)
FCN	0.8201	0.7912	0.8054	0.8496	0.8297	250.8 M	136.3 G	14.7 (68.03)
SegNet	0.8457	0.8068	0.7967	0.8896	0.8550	49 M	286.1 G	21.1 (47.39)
U-Net	0.9021	0.9115	0.9067	0.9021	0.8603	58 M	354.2 G	17.0 (58.82)
Ours	0.9418	0.9329	0.9373	0.9674	0.8972	14.5 M	3.8 G	64.5 (15.50)

**Table 6 sensors-21-02902-t006:** The P, R, F1, ACC, MIoU and FPS of compared methods on GAPs384 dataset (1920×1080 pixels).

Methods	Detection Effect Evaluation Index					
	P	R	F1	ACC	MIoU	FPS (Milliseconds/Image)
FCN	0.6879	0.6179	0.6510	0.8121	0.7257	1.9 (526.31)
SegNet	0.6902	0.6738	0.6819	0.8542	0.7391	2.7 (370.37)
U-Net	0.7092	0.6115	0.6567	0.8861	0.7294	2.2 (454.54)
Ours	0.8980	0.7673	0.8275	0.9554	0.8723	8.2 (121.95)

**Table 7 sensors-21-02902-t007:** The P, R, F1, ACC, MIoU and FPS of compared methods on Crack500 dataset (2000×1500 pixels).

Methods	Detection Effect Evaluation Index					
	P	R	F1	ACC	MIoU	FPS (Milliseconds/Image)
FCN	0.6932	0.6183	0.6536	0.7648	0.7273	1.3 (769.23)
SegNet	0.7001	0.6034	0.6481	0.7912	0.7643	1.8 (555.56)
U-Net	0.7119	0.6083	0.6561	0.8321	0.7991	1.5 (666.67)
Ours	0.8890	0.8521	0.8701	0.9237	0.8753	5.7 (175.44)

**Table 8 sensors-21-02902-t008:** The P, R, F1, ACC, MIoU and FPS of compared methods on AigleRN dataset.

Methods	Detection Effect Evaluation Index					
	P	R	F1	ACC	MIoU	FPS (Milliseconds/Image)
FCN	0.7322	0.8752	0.7973	0.8524	0.7435	9.9 (101.01)
SegNet	0.7685	0.7432	0.7656	0.8652	0.7647	13.7 (72.99)
U-Net	0.8656	0.8016	0.8323	0.8833	0.8213	11.4 (87.72)
Ours	0.8947	0.8283	0.8602	0.9234	0.8745	43.5 (22.99)

**Table 9 sensors-21-02902-t009:** The P, R, F1, ACC, MIoU and FPS of compared methods on CFD dataset.

Methods	Detection Effect Evaluation Index					
	P	R	F1	ACC	MIoU	FPS (Milliseconds/Image)
FCN	0.8228	0.8945	0.8572	0.9210	0.8110	25.4 (39.37)
SegNet	0.8990	0.8947	0.8804	0.9320	0.8160	35.2 (28.41)
U-Net	0.9070	0.8460	0.8710	0.9415	0.8289	29.3 (34.12)
Ours	0.9729	0.9456	0.9590	0.9821	0.8759	111.3 (8.98)

**Table 10 sensors-21-02902-t010:** Error rate of various algorithms.

Crack Images	Error Rate (%)			
	FCN	SegNet	U-Net	CrackDFANet
Crack-only	7.6	4.8	3.2	0.3
Light interference	8.2	9.6	5.2	1.2
Parking line	5.3	3.9	3.7	0.8
Water stains	4.6	4.2	3.1	0.5
Plant disturbances	6.3	4.6	3.3	0.7
Oil pollution	8.3	6.6	5.8	0.6
Shadow	12.26	10.78	9.48	1.0
Mean value	5.75	6.35	4.83	0.73

## Data Availability

Not applicable.

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
