# Peer review of "Automatic Pixel-Level Pavement Crack Recognition Using a Deep Feature Aggregation Segmentation Network with a scSE Attention Mechanism Module"

_sensors, 2021, doi:10.3390/s21092902_

Round 1
Reviewer 1 Report
The paper presents a developed image processing algorithm to detect pavement cracks.
There is good work that has been conducted in this study comparing deep feature aggregation network with the spatial-channel squeeze & excitation (scSE) attention mechanism module, which is called CrackDFANet.
However, there are a few aspects/ comments that need addressing:
- The abstract contained unnecessary details e.g. the name of the data set (GAPs384 dataset, Crack500 dataset, AigleRN dataset and CFD dataset). if the names of these data sets are arbitrary then you do not need to mention their names - just mentioned how many data sets and the main difference between them. This would be more important for the reader to know at this stage.
- The introduction and the paper in general) does not seem to have mentioned what type of pavements have been used in the study to take images. There many types of pavements including concrete, asphalt, ...etc. It is also important to provide more information about the physical properties of this pavement e.g. porosity, roughness,..etc as these properties can be linked with the efficiency of the image analysis and the detection capabilities of the author's software (CrackDFANet).
- It is also important to include some information on the weather condition during the image acquisition phase from the highway.
- Although the paper readability is good it requires extensive improvement...e.g. check the English structure of the sentence in line 105, 116.
Author Response
Dear Editors and Reviewers:
Thank you for your letter and for the comments concerning our manuscript entitled “Automatic pixel-level pavement crack recognition using deep feature aggregation segmentation network with scSE attention mechanism module” (Manuscript Number: 1139282). Those comments are all valuable and very helpful for revising and improving our paper, as well as the important guiding significance to our researches. We have studied comments carefully and have made correction which we hope meet with approval. Revised portion are marked in red in the paper. Please see the attachment.

Reviewer 2 Report
In this study, the authors proposed to employ a new algorithm in detecting cracks on digital images. The algorithm called CrackDFANet is made by the modification and combination of previous algorithms. The CrackDFANet is proven its effectiveness in terms of detection accuracy and speed in comparison with several previous models such as FCN, Segnet, and U-net. This paper is an interesting topic and obtains some outstanding results with the model proposed. Therefore, the manuscript can be published in the journal with its revised form if the following requirement is satisfied. A major revision needs to be performed.
I suggest revising the manuscript and answering the comments/questions listed below. I also would be happy to read the article after revision to make sure that the authors had addressed all reviewer’s comments and suggestions.
One of the main problems in this manuscript is that it is written too long.
- Introduction: I strongly recommend that this section needs to be rewritten to avoid lengthening. The authors should focus only on the main issues related to the contents of this study.
- Section 2:
- Section 2.1: The manuscript focuses on the deep learning model in the identification of crack. You explained in detail five types of traditional methods making this part is much more than section 2.2, even though section 2.2 is for the work related to deep learning. Thus, the authors should emphasize deep learning, not spend so much on the traditional methods. It would be better if only the names of five traditional crack detection methods with the corresponding valuable references are given. Thereafter, sections 2.1 and 2.2 should be merged into only one section 2 - Related work
- Section 2.2: Deep learning-based crack methods have been applied widely. The authors should add more previous works in this section, concentrating on recent studies. I just quickly check and see that so many works can be considered such as:
Bubryur Kim, N. Yuvaraj, K. R. Sri Preethaa & R. Arun Pandian, Surface crack detection using deep learning with shallow CNN architecture for enhanced computation, Neural Computing and Applications, 2021;
Tien-Thinh Le, Van-Hai Nguyen, and Minh Vuong Le, Development of Deep Learning Model for the Recognition of Cracks on Concrete Surfaces, Applied Computational Intelligence and Soft Computing, 2021.
Luqman Ali, Fady Alnajjar, Hamad Al Jassmi, Munkhjargal Gocho, Wasif Khan and M. Adel Serhani, Performance Evaluation of Deep CNN-Based Crack Detection and Localization Techniques for Concrete Structures, Sensors, 2021.
Zirui Wang, Jingjing Yang, Haonan Jiang, and Xueling Fan, CNN Training with Twenty Samples for Crack Detection via Data Augmentation, Sensors, 2020.
Jung Jin Kim, Ah-Ram Kim and Seong-Won Lee, Artificial Neural Network-Based Automated Crack Detection and Analysis for the Inspection of Concrete Structures, Applied Sciences, 2020.
Yung-An Hsieh and Yichang James Tsai, Machine Learning for Crack Detection: Review and Model Performance Comparison, Journal of Computing in Civil Engineering, 2020;
- Lines 320 – 324:
- The authors said you use 3320 crack images from your (1000), GAPs384 (1969), Crack500 (500), AigleRN (269), and CFD datasets (118). You should show that how many images from each dataset to create 3320 samples.
- You said that you employed the LabelMe tool to label the crack images for your data set. I want to know about other datasets. Did you select the original crack images and label them with the LabelMe tool? Or Did you use images labeled in those data set? How is it affect the performance of the model proposed?
- Section 3.2: The authors do not need to show in detail each dataset. I think it is better if four public datasets are presented in one table to avoid lengthening the manuscript. The information, including the number of images, resolution of the image, and references, … should be shown.
- Section 4:
- The authors clearly present the model. I want to know why you utilized the depthwise separable convolution, not spatial separable convolutions or traditional convolution in the backbone.
- You set the number of epochs iteration as 600. Why did you select this number? Could you mention the acceptable value of the loss in the training of the model?
- Sections 5 and 6: It is Ok. The authors clearly show the advantages of CrackDFANet.
- Conclusions:
- Lines 758-766: The authors do not need to explain the model and unnecessary information again.
The authors should show what the main achievements of this study are. Please refer to line 169, “The main contributions of this paper can be summarized as follows”. The first conclusion is not necessary.

Author Response

(The authors gave the same response as above.)

Round 2
Reviewer 1 Report
the revised version is much better and no further comments.
Reviewer 2 Report
Thank you for your response
After reading the revised manuscript and the responses of the authors, I appreciate significant improvements in the revised manuscript. I found that almost all comments and questions were adequately responded, and corrections were made in the manuscript.
I recommend the manuscript for publication with its form now.